# How Predictable is AI progress?

## Abstract

Benchmarks only serve to measure what models are capable of now, not what they *will be* capable of in the future. We find that the ordering of acquired capabilities is remarkably consistent across large populations of AI models, which begs the question of whether one can forecast which specific examples and capabilities future models will solve next. We propose formalizing this problem into a new evaluation task called *progress prediction*: Can we forecast which unsolved problems will be solved next as future models improve? We find that progress is, in fact, predictable. Through an empirical study of hundreds of millions of predictions made by 1,000+ vision models and 1,600+ language models, we find that this predictability is possible due to the consistent order in which models acquire capabilities across architectures, datasets, and modalities.

## 1 Introduction

Benchmarks offer an evaluation of where model capabilities currently stand, but offer little predictive power of where model capabilities will be. Many benchmarks have become obsolete as a result of models becoming more general and transferring to a wider array of tasks. Therefore, the problem of model capability becomes not a problem of 'if', but 'when'. For instance, rather than knowing if a model will create a cure for a disease, knowing when this will happen can be more important as it can be indicative of how to allocate research efforts in the future. Furthermore, 'when' is potentially a more tractable question to answer than 'how'.

To make progress towards this goal, we formalize the idea of AI progress prediction as a task: Given today's models, can we predict which unsolved problems will be solved next as future models improve through increasing scale and algorithmic efficiency? Just as human experts have an intuition for when they are close to solving a problem, model progress is similarly predictable.

The problem described above can be viewed as an ordering problem where prediction is only possible if there exists a consistent ordering across a broad scope of models regardless of factors like architecture, data, and modality. We first analyze the patterns of correct and incorrect inferences from a large and diverse population of models on a given benchmark. We find that this ordering is remarkably consistent: some examples are reliably solved by a greater number of models than others, regardless of model architecture or training data. This stability reveals a capability frontier, a boundary separating the set of examples solvable by a model of a given accuracy from those that are not.

The task of progress prediction, therefore, is to predict where this frontier will advance. For any given model $M$ and dataset $D$, we consider the set of examples $N^{M,D}$ that the model currently fails to solve. The goal is to identify the subset of examples from $N^{M,D}$ that appear earliest in the population consensus ordering, as these represent the next capabilities the model is most likely to acquire (see Figure 1).

Our main contributions are:

1. Formalizing *progress prediction* as a task and proposing a metric to evaluate prediction quality over this task.

2. Providing an initial baseline for the community of several model metrics and their progress prediction scores.

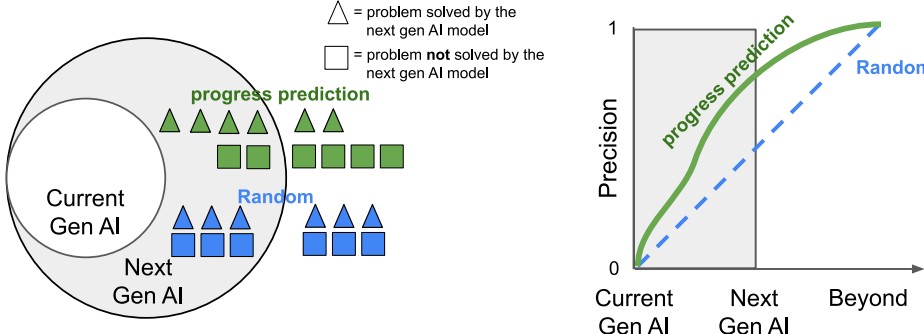

Figure 1: We treat predicting AI model progress as a binary classification problem. For the purposes of illustrating our method, let's assume we know which problems will be solved by a next generation AI model. The classification task is: when the next generation AI model is released, which data points will it be able to solve and which will it not be able to solve. We can create a metric around this, simply what next solvable predictions are inside the set of problems solved by current models and which are not. Triangles are the problems that are solved by a hypothetical next generation AI model. Squares are the problems that are not solved by a hypothetical next generation AI model.

3. Designing the *Prediction Order Coherence (POC)* metric, for measuring population example correctness ordering and applying it to large populations of vision (1,000+) and language models (1,600+) across many popular benchmark tasks.

4. Discovering correlations between population example difficulty ordering and learning order during model training.

To support these contributions, we first establish there is roughly a frontier of problems that only a model of sufficient accuracy can solve. Then we demonstrate that the next unsolved problems on which the frontier advances can be predicted.

## 2 RELATED WORK

We study example difficulty through cross-model agreement. Hacohen et al. (2020) show that different neural networks broadly agree on which images are easy or hard to classify. Prior work on training–dynamics also proposes a stable instance-level structure to learning: Toneva et al. (2019) identifies "unforgettable" and frequently forgotten examples, while Swayamdipta et al. (2020) maps data into easy/ambiguous/hard regions using confidence and variability over epochs (stable across random initializations). Model-based proxies such as the *c-score* (Jiang et al., 2021), *prediction depth* (Baldock et al., 2021), and *VoG* (Agarwal et al., 2022) operationalize difficulty signals from training behavior. Human-grounded difficulty via Minimum Viewing Time (MVT) (Mayo et al., 2023) also provides an orthogonal, perception-based scale.

A related line of work uses example signal for optimization during training. Classical curriculum learning (Bengio et al., 2009; Hacohen & Weinshall, 2019) and its modern variants (e.g., *Data Diet* (Paul et al., 2021), *Curriculum by Smoothing* (Sinha et al., 2020)) rank or stage examples from easier/near-correct to harder, while targeted data selection (Xia et al., 2024; Bi et al., 2025; Sorscher et al., 2022) prioritizes influential data to reduce redundant training. Our focus differs in that we aggregate many models and use such signals for predicting the future of the AI frontier, rather than optimizing any currently viable task.

Other forecasting works have aggregated performance as a function of scale or time (Hestness et al., 2017; Kwa et al., 2025; Sevilla et al., 2022), but such works provide *macro-level* trends. Our study proposes the first cross-domain technique for predicting *micro-level* order in learning across models.

# 3  MEASURING THE ORDER CONSISTENCY OF POPULATIONS OF MODELS

## 3.1  DATA COLLECTION

We collected model predictions across language, vision, and protein domains. For language models, we leveraged the HuggingFace Open LLM Leaderboard evaluations, which provides evaluation results for over 1,600 language models across 5 challenging datasets:

1. **Big Bench Hard (BBH)** (Suzgun et al., 2022a; bench authors, 2023): A subset of 23 challenging tasks from BigBench that cover algorithmic reasoning, language understanding, and world knowledge. The tasks include: Sports Understanding, Tracking Shuffled Objects, Navigate, Snarks, Date Understanding, Reasoning about Colored Objects, Object Counting, Logical Deduction (with Three, Five, and Seven Objects), Geometric Shapes, Web of Lies, Movie Recommendation, Salient Translation Error Detection, Disambiguation QA, Temporal Sequences, Hyperbaton, Causal Judgement, Formal Fallacies, Ruin Names, Penguins in a Table, and Boolean Expressions. Each task has a specific number of choices (ranging from 2 to 19).
2. **MATH** (Hendrycks et al., 2021b): We used the Level 5 (most difficult) subset of MATH questions, which contain high-school competition-level mathematics problems. We evaluated models by giving the model 4 examples and then checking for a latex boxed solution that either exactly matched or reduced to the correct solution.
3. **Graduate-Level Google-Proof Q&A Benchmark (GPQA)** (Rein et al., 2024): A challenging knowledge dataset with questions crafted by PhD-level domain experts in fields like biology, physics, and chemistry. We evaluated the models 0-shot and for each question there were 4 multiple-choice options.
4. **Multistep Soft Reasoning (MUSR)** (Sprague et al., 2024): A dataset of algorithmically generated complex problems requiring integration of reasoning with long-range context parsing. We evaluated three subtasks: Murder Mysteries (0-shot, 2 choices), Object Placement (0-shot, 5 choices), and Team Allocation (0-shot, 3 choices).
5. **Massive Multitask Language Understanding Pro (MMLU-Pro)** (Hendrycks et al., 2021a; Wang et al., 2024): A refined version of MMLU with 10 choices per question (instead of 4) and expert-reviewed content to reduce noise. Models were provided with 5 examples.

For all language model evaluations, we extracted binary correctness judgments (correct/incorrect) for each model-example pair by evaluating model response against ground truth answers ourselves. This resulted in a binary prediction matrix for each task, visualized in Figure 3.

For vision models, we evaluated 1,000+ models from the timm repository (Wightman, 2019) across three diverse computer vision benchmarks: ImageNet (Krizhevsky et al., 2012), ObjectNet (Barbu et al., 2019), and LAIONet (Shirali & Hardt, 2023). ImageNet represents a standard object recognition task, ObjectNet tests out of distribution robustness controlling for viewpoint, rotation, and background, and LAIONet tests generalization to an ImageNet like subset of LAION400m, offering a larger scale of over 500k images that could in the future be used along with population difficulty information for training. We conducted inference using each model's standard configuration and classified a prediction as correct if the ground truth class appeared in the model's top-1 prediction.

To ensure consistency in our analysis, we filtered both language and vision model populations to include only models with accuracy above 20% on their respective benchmarks, eliminating models that perform near random chance. This resulted in a final dataset containing hundreds of millions of individual model predictions, creating what we believe is the largest model population prediction analysis to date.

Additionally, we trained 100 ResNet50 models on ImageNet, each with a different random seed, and evaluated them on the ImageNet validation set to provide a baseline level of prediction agreement when controlling for model architecture, training data, and all other training parameters.

For protein models, we included evaluations from ProteinGym benchmarks (Notin et al., 2023), focusing on DMS (Deep Mutational Scanning) datasets for both substitution and indel (insertion-deletion) mutations. Unlike the vision and language benchmarks, the success of a model's predictions are measured in a continuous metric. To fit our framework, we set a threshold to binarize these

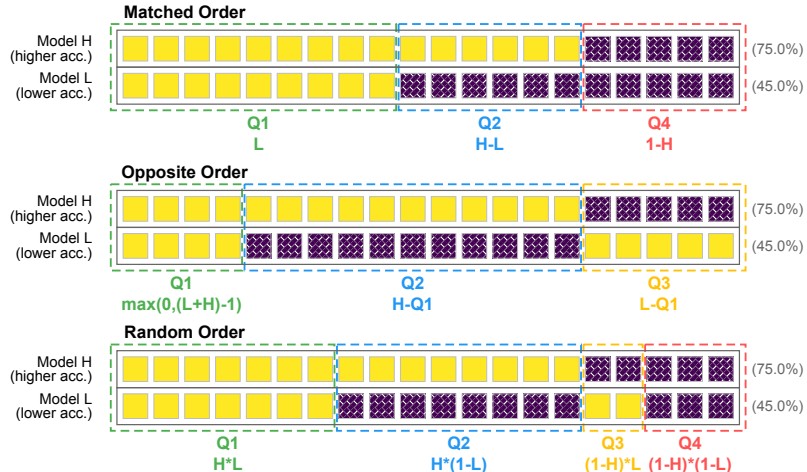

Figure 2: Possible prediction patterns between two models of different accuracy: **Matched order:** Correct predictions maximally overlap between models. A better model gets all the examples correct that a worse model gets correct plus some. **Opposite order:** Correct predictions minimally overlap between models. **Random order:** Predictions made by each model are independent.

predictions. These benchmarks test models' ability to predict the impact of mutations on protein function, providing another domain to observe the degree to which population ordering holds true.

### 3.2 METRICS

To quantify the extent to which models follow a consistent ordering in their predictions, we developed the Prediction Order Coherence (POC) metric.

#### 3.2.1 PREDICTION ORDER COHERENCE (POC)

Prediction Order Coherence measures the degree to which the predictions of higher-accuracy models subsume those of lower-accuracy models. Given a binary prediction matrix $M \in \{0,1\}^{n \times m}$ where rows represent models (sorted by row sums from greatest to least) and columns represent examples (sorted by column sums from greatest to least), with values indicating correct (1) predictions or incorrect (0) predictions as seen in Figure 3, POC quantifies whether higher-accuracy models consistently solve the problems that lower-accuracy models can solve, plus additional ones.

There are 4 possible combinations of binary correctness patterns between a pair of 2 models. We sort model pairs such that the overall higher accuracy model comes first and define four quadrants of the binary correctness patterns: Q1) correct-correct, Q2) correct-incorrect, Q3) incorrect-correct, and Q4) incorrect-incorrect. These are labeled for each prediction pattern in Figure 2. POC measures the counts of Q2 for all pairs of models in a population of models, normalizes such that 1 is a matched order and 0 is an opposite ordering.

$$Q2_{\text{model pop.}}(M) = \sum_{1 \leq i < j \leq n} \sum_{k=1}^{m} \mathbf{1}\{M_{i,k} = 1,\ M_{j,k} = 0\} \tag{1}$$

The POC score is computed by normalizing between the two extremes:

$$\text{POC} = \frac{Q2_{\text{model pop.}} - Q2_{\text{opposite}}}{Q2_{\text{matched}} - Q2_{\text{opposite}}} \tag{2}$$

Where:

- $Q2_{\text{model pop.}}$ is the sum of the counts where the higher-accuracy model is correct and the lower-accuracy model is incorrect across all model pairs and examples.

- $Q2_{\mathrm{matched}}$ represents the theoretical maximum (perfect ordering) where higher-accuracy models perfectly subsume lower-accuracy models (derivation: Appendix A.5).
- $Q2_{\mathrm{opposite}}$ represents the theoretical minimum (adversarial ordering) where higher-accuracy models systematically get different examples correct than lower-accuracy models (derivation: Appendix A.3).

A POC score of 1.0 indicates perfect ordering (the matched order case in Figure 2), while a score of 0.0 indicates a complete reversal of the expected ordering (the opposite order case). Random prediction patterns typically yield POC scores near zero, but this depends on the average accuracy of the models in the population.

## 4    THE NEXT SOLVABLE EXAMPLE PREDICTION TASK

Here we precisely formulate our framework for measuring a method's ability to predict what as-of-yet unsolved examples will be solvable in the near future.

First, how can we "know" what examples a model will be able to solve next if it improves its overall accuracy? We use the population difficulty order derived from a large population of models spanning a range of accuracies and back-test: selecting a lower accuracy model and asking if it can somehow predict the set of examples that a future slightly more accurate model would solve. We look into the future for increasing horizons from 0% to 100% of yet unsolved examples. By taking the Area Under the Curve (AUC) between how accurately we can predict the next percent of examples compared to random chance, we get an AUC metric that captures how well we can predict the future, where random predictions get an AUC of $0.5$.

For every model $M$ and dataset $D$, there are some examples in $D$ which $M$ can predict correctly and some examples which it cannot predict correctly (we are not interested in solved datasets $D$ where $M$ solves every example correctly).

For the set of examples that $M$ gets incorrect, there is an underlying population order of how "difficult" each example is. We ask the simple question, can model $M$ predict the easiest K examples out of all the examples that it couldn't solve?

For every model $M$ we define $N^{M,D}$ as the set of examples in dataset $D$ that model $M$ is unable to solve. Then for every value of $K$ where $K \in \{1, ..., |N^{M,D}|\}$ we can calculate the percentage set intersection (or precision) between the model's predicted set and the true set of easiest $K$ examples from $N^{M,D}$. If we randomly pick $K$ examples from the set $N^{M,D}$ then the precision in expectation would trivially be $\frac{K}{N^{M,D}}$.

In Figure 6, we plot the value of $\frac{K}{N^{M,D}} \in [0, 1]$ against the set intersection percent (or precision) at K. We repeat this for 37 vision models (Figures 6a and 6b) and 26 LLMs (Figures 6c and 6d listed in Appendix A.7 selected to cover a range of overall model accuracies.

The two figures in the left column (Figures 6a and 6c) rely on knowing the ground truth labels for each example to extract a future prediction while the two figures in the right column (Figures 6b and 6d) do not rely on knowing the ground truth labels and merely use other simple statistics of the overall output logits.

There are significant implications between the two columns, while the left column achieves a higher AUC, the right column can make future predictions about each sample even if neither humans nor machines know the correct answer to it, the models are still able to consistently make progress predictions better than random.

## 5    RESULTS

### 5.1    OBSERVED POPULATION PREDICTION PATTERNS

The aggregated prediction results from the model populations are shown in figure 3. The matrices show a far more matched than random ordering in which higher accuracy models tend to subsume the successes of lower accuracy models across all modalities and tasks, though it is most visible for vision models.

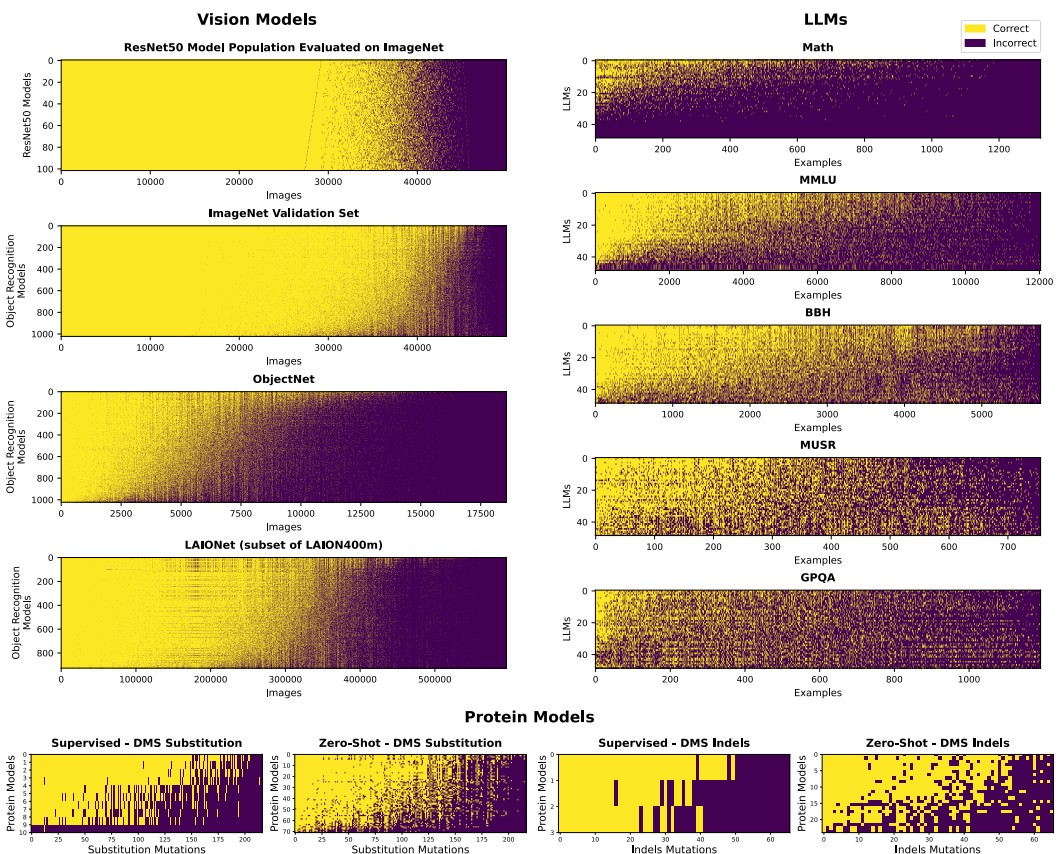

Figure 3: Model population predictions on several vision Krizhevsky et al. (2012); Barbu et al. (2019); Shirali & Hardt (2023), language Hendrycks et al. (2021b); Wang et al. (2024); Suzgun et al. (2022b); Sprague et al. (2024); Rein et al. (2024) and protein Notin et al. (2023) benchmarks. The rows in each of the bars are models, the columns are dataset examples, and each cell is a model prediction scored as correct (yellow) or incorrect (purple). For LLMs we visualize a representative set of models, not including models of all scales and variants of the same family.

We measure the degree of this ordering using the Prediction Order Coherence metric as seen in Figure 6. Vision models on object recognition tasks exhibit a higher POC while LLMs exhibit lower POC values. We hypothesize that tasks that require less trivial knowledge and more compositional skill will exhibit a higher POC. This degree of observed ordering is what enables progress prediction, predicting where the frontier will move next.

For ImageNet, the most matched ordered task according to POC, we visualize several example images across the range of population rankings in 7.

## 5.2 NEXT SOLVABLE EXAMPLE AUC

For every model $M$ and dataset $D$ we generate a plot of $\frac{K}{N^{M,D}} \in [0, 1]$ vs the set intersection percent (or precision) at $K$ (also $\in [0, 1]$) which represents the next $K$ examples that it thinks will be solved. We repeat this for all the models and average the lines to get a single mean line representing the population average. We then calculate the AUC of this plot. A perfect predictor would obtain an AUC of $1$ while a completely random predictor would get $0.5$.

In Figure 6 we report Vision Models and LLMs separately as they operate on different datasets. We additionally separate predictions that rely on us knowing the examples ground truth label from predictions that don't rely on such information. For each individual model, we use simple metrics (like the ground logit or the entropy of the logits) to output a prediction of difficulty. A comprehensive list of metrics are listed in Appendix A.2

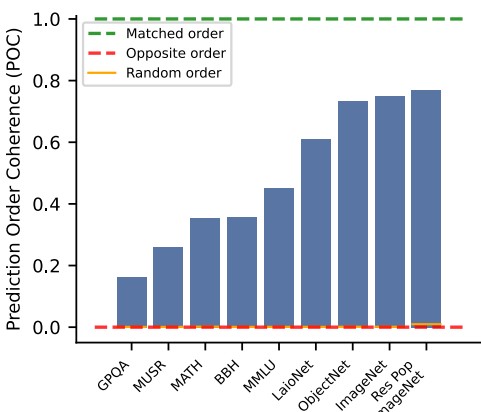

Prediction Order Coherence (POC)

Figure 4: Agreement between model predictions and population patterns across different tasks. Green dashed line indicates perfect agreement (matched order), blue dashed line represents maximal disagreement (opposite order), and orange line shows agreement with a simulated random ordering of predictions.

Table 1: Spearman correlation values ($\rho$) between population prediction order and difficulty metrics.

| Dataset | Metric | $\rho$ |
|---|---|---|
| ImageNet | ResNet50 cscores | **0.794** |
| ImageNet (subset) | cscore | 0.670 |
| | dscore (MVT) | 0.318 |
| | adversarial epsilon | 0.441 |
| | prediction depth | 0.198 |
| ObjectNet | ResNet50 cscores | **0.748** |
| ObjectNet (subset) | cscore | 0.736 |
| | dscore (MVT) | 0.494 |
| | adversarial epsilon | 0.493 |
| | prediction depth | 0.169 |

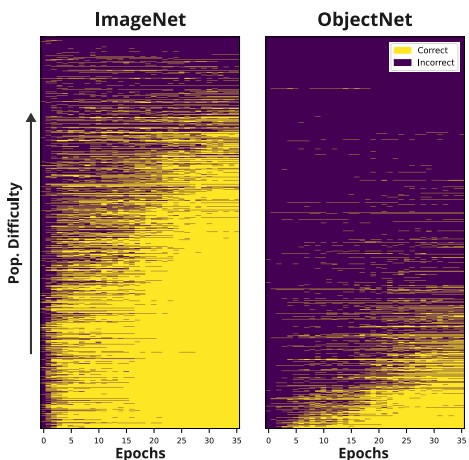

Figure 5: Predictions made by a ResNet50 at each training epoch sorted by population difficulty for both the ImageNet validation set (left) and the ObjectNet test set (right).

Our findings show that consistently across almost all models and multiple modalities, models are able to predict progress on the subset of examples that they cannot solve yet ($0.642$ AUC and $0.597$ AUC compared to random $0.5$). To highlight the strength of our findings, model-based metrics can predict progress even when knowledge of the ground truth label is not used. Figure 6b and Figure 6d show that we can consistently, above random, predict progress without having access to ground truth labels ($0.545$ for vision and $0.550$ for language, random is $0.5$).

## 5.3 CORRELATION WITH EXISTING DIFFICULTY METRICS

To understand how our population-based difficulty ordering relates to existing notions of example difficulty, we compared our ordering with several established metrics, summarized in Table 1.

For ImageNet and ObjectNet, we examined correlations with:

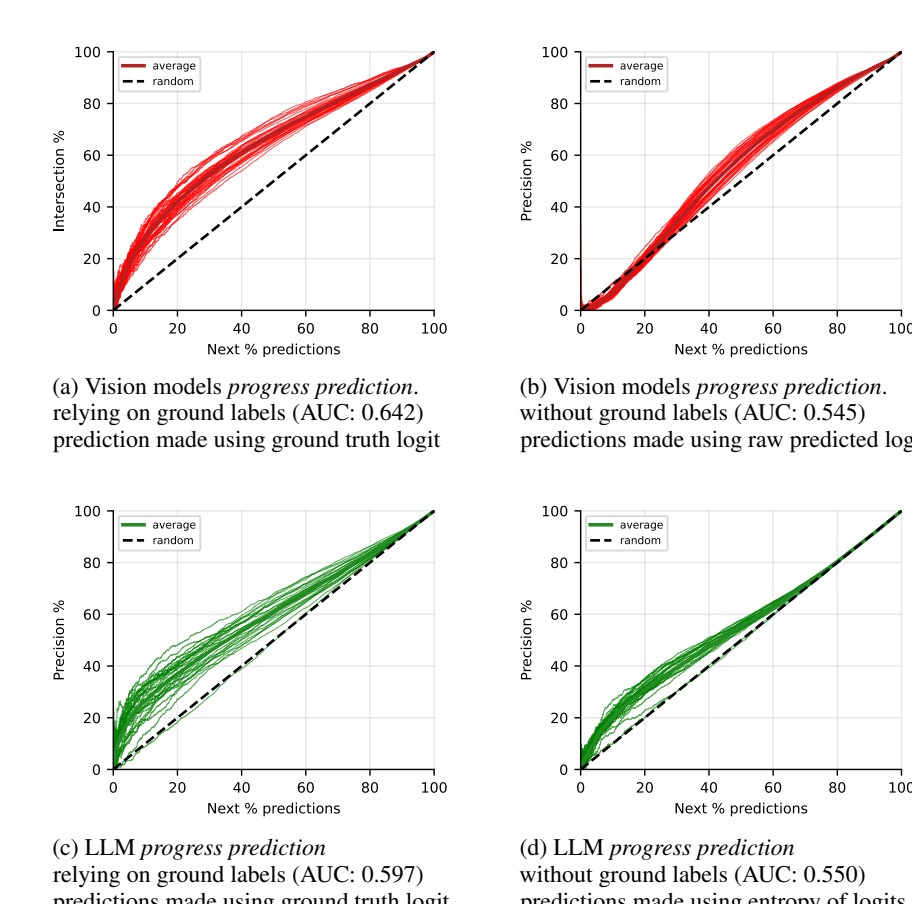

(a) Vision models *progress prediction*.
relying on ground labels (AUC: 0.642)
prediction made using ground truth logit

(b) Vision models *progress prediction*.
without ground labels (AUC: 0.545)
predictions made using raw predicted logit

(c) LLM *progress prediction*
relying on ground labels (AUC: 0.597)
predictions made using ground truth logit

(d) LLM *progress prediction*
without ground labels (AUC: 0.550)
predictions made using entropy of logits

Figure 6: Comparison of different progress prediction methods for next percent prediction. These 4 metrics used to make a progress prediction, ground truth logit confidence (6a), predicted logit confidence (6b), ground truth logit confidence (6c), and entropy of logits (6d). 26 LLMs and 37 vision models were used for these experiments.

- **ResNet50 c-scores**(Jiang et al., 2021): We used a time learning proxy for cscore, a model-based difficulty metric that measures how early during training a model learns to classify an image correctly. Images with higher c-scores are learned earlier in training and are considered easier.
- **d-scores (MVT)** (Mayo et al., 2023): Minimum Viewing Time required by humans to recognize an object in an image. This is an objective, model-free metric where images that humans can recognize with brief flashes are easy, while those requiring seconds of viewing are hard.
- **Adversarial epsilon** (Mayo et al., 2022): The minimum perturbation magnitude required to cause a model to misclassify an image, where larger values indicate greater robustness.
- **Prediction depth** (Baldock et al., 2021): The earliest layer in a neural network at which the model's prediction matches its final output, indicating where in the network the decision stabilizes.

Our analysis revealed interesting correlations between population-based difficulty ordering and these existing metrics (Table 1). The strongest correlation was with ResNet50 c-scores (Spearman $\rho = 0.794$ for ImageNet and $\rho = 0.748$ for ObjectNet). This high degree of correlation suggests that we have effectively been training the same big model over the past decade-long history of training deep learning models for object recognition, each time advancing the frontier set of solvable problems a bit. This pattern is visualized in 5.

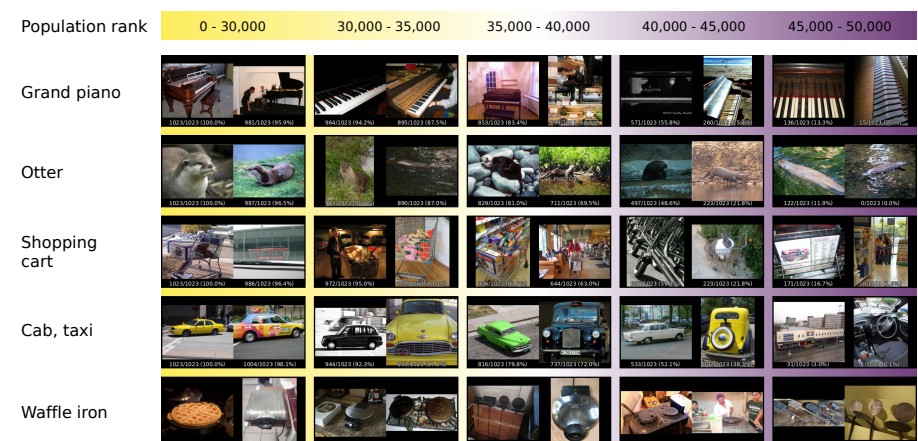

Figure 7: ImageNet images of different population order difficulties

We found moderate positive correlations with adversarial epsilon ($\rho = 0.441$ for ImageNet and $\rho = 0.493$ for ObjectNet), suggesting that examples that are more difficult according to the population ordering tend to also be more vulnerable to adversarial attacks.

We also observed a positive correlation with d-scores (MVT) ($\rho = 0.318$ for ImageNet and $\rho = 0.494$ for ObjectNet). The MVT metric measures how much viewing time humans need to recognize an object in an image. This degree of correlation makes population difficulty order one of the best predictors of human difficulty as measured by MVT compared to the other metrics evaluated in (Mayo et al., 2023).

## 6 DISCUSSION

As AI systems approach solving high-value problems in science, medicine, and other domains, predicting not just if, but when specific problems will be solved becomes an important problem for prioritizing research and reasoning about the future.

We pose the problem of progress prediction: forecasting which currently unsolved examples will be solved next as AI models improve. We offer baseline methods for predicting future solvable examples using model confidence (Figure 6), with additional metrics explored in the appendix. These simple metrics consistently perform above chance, achieving AUCs of 0.642 (vision) and 0.597 (language) when using ground truth labels.

This above chance predictability is enabled by the high degree of order consistency across modalities and tasks that we observe in model predictions across modalities and datasets, quantified by our POC metric.

We invite the research community to develop more sophisticated progress prediction methods and apply our evaluation framework to measure their success. The consistent ordering we observe suggests that more accurate predictions are possible, potentially enabling researchers to anticipate and prepare for emerging AI capabilities before they arrive.

## 7 LIMITATIONS AND COMPUTE RESOURCES

While our work relies on datasets where we observe consistent orderings across a model population, there are datasets where this ordering is weaker. These datasets tend to be ones where examples do not share common skill dependencies such as trivia problems. In contrast, we find a more matched ordering present in compositional skills like reasoning and mathematics. 4 H100 GPUs were used for evaluating the vision model populations. We are constrained by existing models and datasets in what model evaluations we are able to study.

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

## A  APPENDIX

### A.1  FUTURE PREDICTIONS AS BINARY CLASSIFICATION

In this section, we treat the task of predicting the future similar to Figure 6 but as a binary classification problem where we only attempt to predict the next $X\%$ of examples.

For each $X \in \{5, 10, 20, 50\}$ we have 4 plots (thus 16 in total) in Figures 8,9, and 10,11. The two plots in the top row are generated using the 37 vision models listed in Appendix A.7 while the two plots in the bottom row are generated using the 26 LLMs also mentioned in Appendix A.7. There is a significant difference between the left column and the right column; While the left column achieves a higher AUC, it does depend on us knowing the ground truth labels of each example (as the ground softmax is used for the prediction) while the right column does not depend on the ground truth label at all, merely on statistics of the outputted set of logits. The significance of this is that the right column tells us that even if neither humans nor the models knew the correct answer, the models are still able to know that the examples are approaching being solved (consistently better than random).

When we refer to "logits total variation" for LLMs we specifically take the logits of each of the $n$ possible multiple choices answers then computes $\|L_i - \frac{1}{n}\|_1$, this metric was chosen as it had achieved the best progress prediction without knowing the ground label. For a comprehensive list of every metric tried, we list all of them in Appendix A.2 along with the AUC for the $5\%$ future prediction task (similar to Figure 8).

### A.2  COMPREHENSIVE LIST OF FEATURES AND ROCS

Here we list a comprehensive list of many metric we tried along with the AUC ROC for the $5\%$ future prediction task (similar to Figure 8). The metrics are for both vision models and LLMs, and every metric either relies on or does not rely on the ground label. The four tables are Table 2, Table 3, Table 4, and Table 5

Table 2: AUC ROC of metrics from vision models that rely on the ground label

| Value | AUC ROC |
|---|---|
| Targets Softmax | 0.793 |
| Targets Logits | 0.725 |

### A.3  DERIVATION OF $Q2_{\text{OPPOSITE}}$

Here we derive how we obtain $Q2_{\text{opposite}}$ representing the theoretical minimum (opposite ordering), where higher-accuracy models systematically get different examples correct than lower-accuracy models. We define that the theoretical minimum ordering (or opposite ordering) is when the sum of the quantity $Q_3$ from Figure 2 is maximized across all pairs of models.

Let $M \in \{0, 1\}^{n \times m}$ be a binary matrix with rows ($i = 1, \ldots, n$) and columns ($k = 1, \ldots, m$) where $M_{ik} = 1$ implies that model $i$ predicts sample $k$ correctly. Then we fix the row sums of $M$ to match our empirical real model population accuracies such that the sum of the $i$-th row equals $L(i)$ where $L(i)$ is the number of samples that model $i$ gets correctly. Assume the rows of $M$ are ordered

Table 3: AUC ROC of metrics from vision models that do not rely on the ground label

| Value | AUC ROC |
|---|---|
| Logits Cr 2 | 0.560 |
| Predicted Softmax | 0.557 |
| Logits Cr 3 | 0.537 |
| Logits Mode Value | 0.534 |
| Logits Mean | 0.531 |
| Logits Std | 0.529 |
| Logits Variance | 0.529 |
| Logits Renyi Q3 | 0.528 |
| Logits Renyi Q2 Entropy | 0.526 |
| Logits Participation Ratio | 0.526 |
| Logits Simpson Index | 0.526 |
| Logits Hill Number Q2 | 0.526 |
| Logits Renyi Q2 | 0.526 |
| Logits Gini Impurity | 0.526 |
| Logits Hhi | 0.526 |
| Logits Tsallis Q2 Entropy | 0.526 |
| Logits Shannon Entropy | 0.525 |
| Logits Kl To Uniform | 0.524 |
| Logits Cr 1 | 0.524 |
| Logits Max Prob | 0.524 |
| Logits Min Entropy | 0.524 |
| Logits Mode Prob | 0.524 |
| Logits Hellinger Distance To Uniform | 0.523 |
| Logits Js Divergence To Uniform | 0.523 |
| Logits Skewness | 0.523 |
| Logits Kurtosis Excess | 0.523 |
| Logits Total Variation To Uniform | 0.523 |
| Predicted Logits | 0.509 |
| Logits Hill Number Q1 | 0.505 |
| Logits Perplexity | 0.505 |
| Logits Cr 5 | 0.504 |
| Logits Iqr | 0.501 |
| Logits Q25 | 0.501 |
| Logits Median | 0.500 |
| Logits Q75 | 0.500 |
| Logits Min Prob | 0.500 |
| Logits Min Value | 0.500 |
| Logits Max Value | 0.500 |
| Logits Hill Number Q0 | 0.500 |

Table 4: AUC ROC of metrics from LLMs that rely on the ground label

| Value | AUC ROC |
|---|---|
| Target Logit | 0.708 |
| Target Softmaxed | 0.636 |

Table 5: AUC ROC of metrics from LLMs that do not rely on the ground label

| Value | AUC ROC |
|---|---|
| Logits Cr 2 | 0.688 |
| Logits Cr 3 | 0.686 |
| Softmaxed Cr 5 | 0.678 |
| Logits Renyi Q3 | 0.677 |
| Logits Participation Ratio | 0.677 |
| Logits Hill Number Q2 | 0.677 |
| Logits Hhi | 0.677 |
| Logits Simpson Index | 0.677 |
| Logits Renyi Q2 Entropy | 0.677 |
| Logits Renyi Q2 | 0.677 |
| Logits Tsallis Q2 Entropy | 0.677 |
| Logits Gini Impurity | 0.677 |
| Logits Total Variation To Uniform | 0.677 |
| Logits Shannon Entropy | 0.676 |
| Logits Perplexity | 0.676 |
| Logits Hill Number Q1 | 0.676 |
| Logits Kl To Uniform | 0.676 |
| Logits Js Divergence To Uniform | 0.675 |
| Logits Hellinger Distance To Uniform | 0.675 |
| Softmaxed Cr 3 | 0.673 |
| Logits Cr 5 | 0.670 |
| Logits Mode Prob | 0.668 |
| Logits Cr 1 | 0.668 |
| Logits Max Prob | 0.668 |
| Logits Min Entropy | 0.668 |
| Softmaxed Min Prob | 0.666 |
| Softmaxed Cr 2 | 0.659 |
| Softmaxed Hellinger Distance To Uniform | 0.648 |
| Softmaxed Js Divergence To Uniform | 0.640 |
| Softmaxed Total Variation To Uniform | 0.636 |
| Softmaxed Kl To Uniform | 0.629 |
| Softmaxed Perplexity | 0.629 |
| Softmaxed Shannon Entropy | 0.629 |
| Softmaxed Hill Number Q1 | 0.629 |
| Pred Logit | 0.627 |
| Logits Min Prob | 0.625 |
| Softmaxed Hill Number Q2 | 0.613 |
| Softmaxed Hhi | 0.613 |
| Softmaxed Gini Impurity | 0.613 |
| Softmaxed Participation Ratio | 0.613 |
| Softmaxed Simpson Index | 0.613 |
| Softmaxed Tsallis Q2 Entropy | 0.613 |
| Softmaxed Renyi Q2 | 0.613 |
| Softmaxed Renyi Q2 Entropy | 0.613 |
| Softmaxed Renyi Q3 | 0.607 |
| Pred Softmaxed | 0.602 |
| Softmaxed Cr 1 | 0.599 |
| Softmaxed Mode Prob | 0.599 |
| Softmaxed Max Prob | 0.599 |
| Softmaxed Min Entropy | 0.599 |
| Softmaxed Variance | 0.586 |
| Softmaxed Std | 0.586 |
| Softmaxed Iqr | 0.558 |
| Logits Kurtosis Excess | 0.557 |
| Softmaxed Skewness | 0.543 |
| Logits Mode Value | 0.540 |
| Softmaxed Mode Value | 0.540 |
| Softmaxed Kurtosis Excess | 0.540 |
| Softmaxed Q25 | 0.539 |
| Logits Q75 | 0.538 |
| Logits Std | 0.533 |
| Logits Variance | 0.533 |
| Logits Mean | 0.529 |
| Softmaxed Median | 0.526 |
| Logits Iqr | 0.525 |
| Logits Skewness | 0.523 |
| Logits Q25 | 0.515 |
| Softmaxed Mean | 0.514 |
| Softmaxed Q75 | 0.502 |
| Logits Median | 0.502 |
| Logits Hill Number Q0 | 0.500 |
| Logits Min Value | 0.500 |
| Logits Max Value | 0.500 |
| Softmaxed Min Value | 0.500 |
| Softmaxed Hill Number Q0 | 0.500 |
| Softmaxed Max Value | 0.500 |

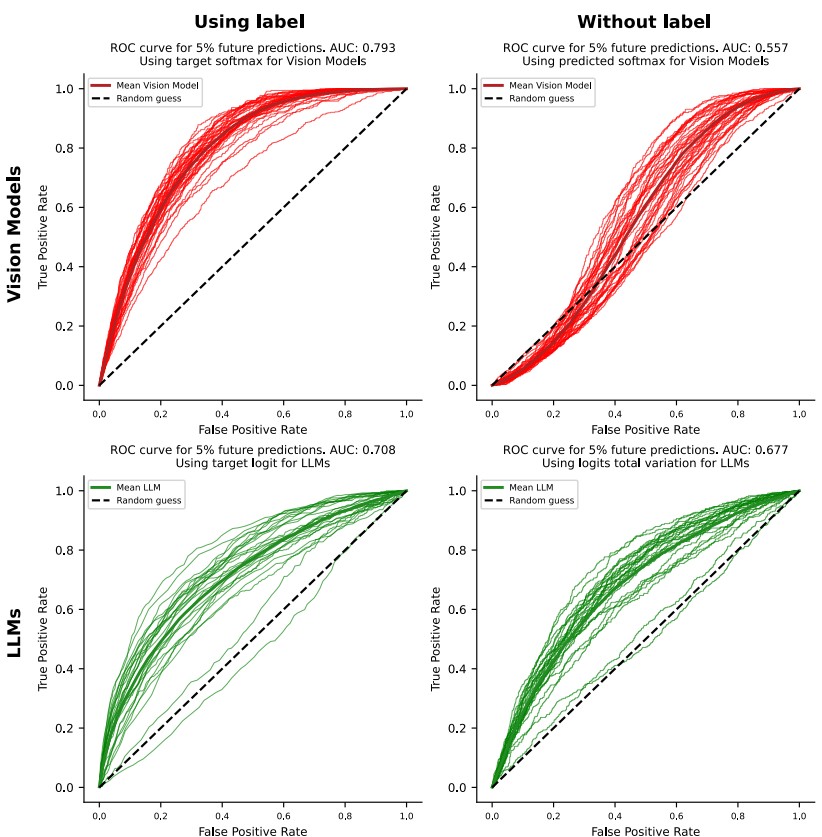

Figure 8: Here we only attempt to predict the next 5% ahead of each model's capabilities. In all plots, we again see the consistent trend that almost all models are capable of predicting the near future beyond what they are capable of solving.

such that the associated accuracies satisfy $L(1) \geq L(2) \geq \cdots \geq L(n)$. The exact sorting of rows isn't required, but for the purposes of this derivation, it makes the equations simpler and easier to follow if the rows are sorted in non-increasing order. We then define the function $Q_3$ (eq. 3):

$$Q_3(i,j) := |\{k \in \{1, \ldots, m\} : M_{i,k} = 0, \ M_{j,k} = 1\}| \tag{3}$$

Where $(1 \leq i < j \leq n)$ (i.e., the number of samples that a lower accuracy model $j$ gets right, while the higher accuracy model $i$ gets wrong). Then we note the simple relation:

$$L(j) = \sum_{k=1}^{m} \mathbf{1}\{M_{j,k} = 1\} = \sum_{k=1}^{m} \mathbf{1}\{M_{j,k} = 1, \ M_{i,k} = 0\} + \sum_{k=1}^{m} \mathbf{1}\{M_{j,k} = 1, \ M_{i,k} = 1\} \tag{4}$$

$$\sum_{k=1}^{m} \mathbf{1}\{M_{j,k} = 1, \ M_{i,k} = 0\} = L(j) - \sum_{k=1}^{m} \mathbf{1}\{M_{j,k} = 1, \ M_{i,k} = 1\} \tag{5}$$

Then we define $\mathrm{DIS}(M)$ (eq. 6)

$$\mathrm{DIS}(M) = \sum_{1 \leq i < j \leq n} Q_3(i,j). \tag{6}$$

Where $\mathrm{DIS}(M)$ is what we want to maximize to obtain $Q2_{\mathrm{opposite}}$ representing our opposite ordering. Given a matrix $M$, it is trivial to algorithmically calculate $\mathrm{DIS}(M)$, therefore, for our main goal of obtaining $Q2_{\mathrm{opposite}}$ is to obtain the matrix $M$ that maximizes $\mathrm{DIS}(M)$.

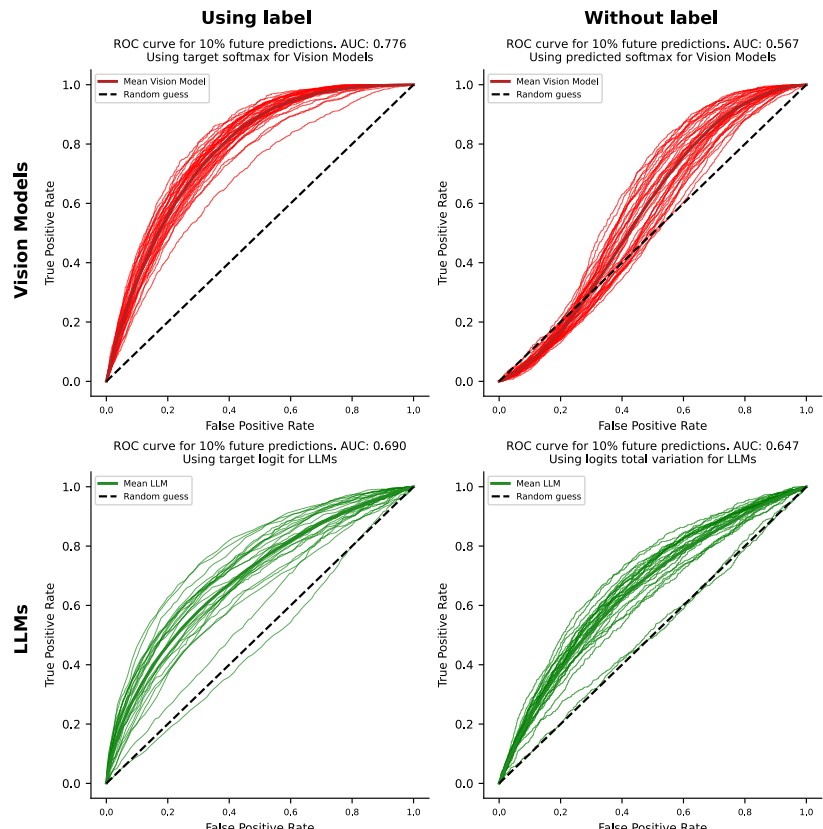

Figure 9: Similar to Figure 8 but this time predicting the next 10% ahead of each model's capabilities.

$$\underset{M \in \{0,1\}^{n \times m}}{\arg\max} \quad \text{DIS}(M) \quad \text{s.t.} \ \sum_{k=1}^{m} M_{ik} = L(i) \ \forall i. \tag{7}$$

Thus, we derive the following (the row-sum constraint is omitted for brevity):

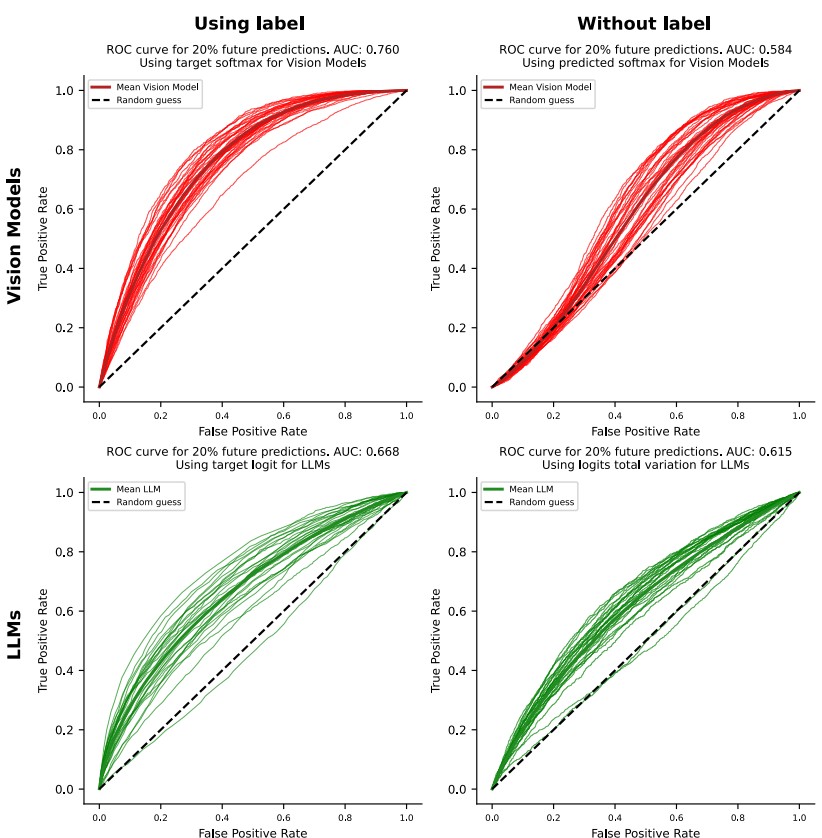

Figure 10: Similar to Figure 8 but this time predicting the next 20% ahead of each model's capabilities.

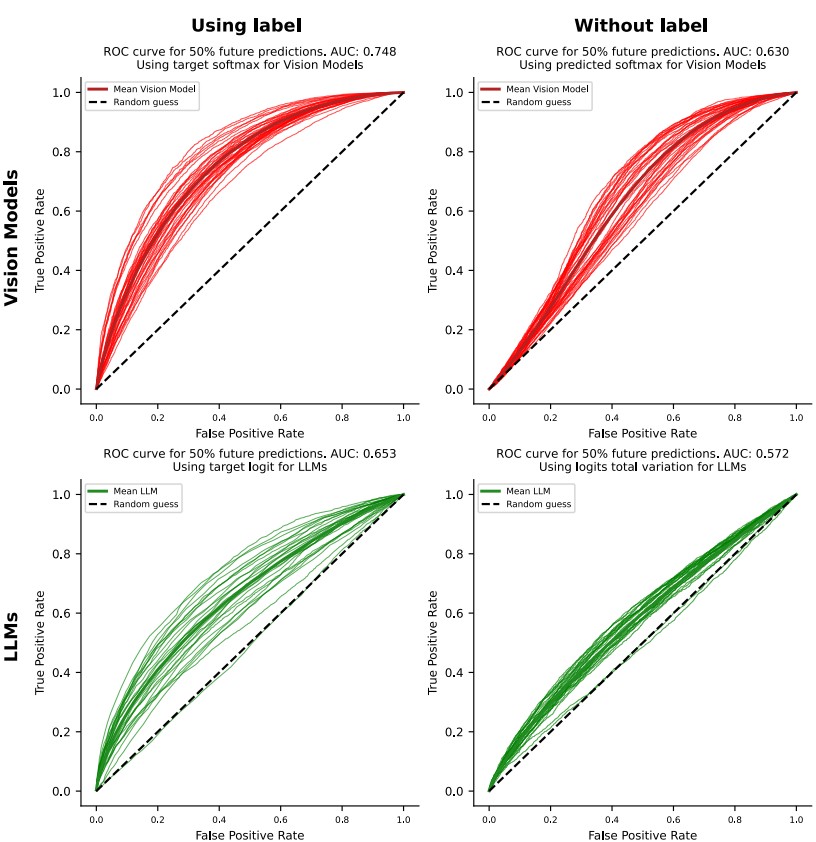

Figure 11: Similar to Figure 8 but this time predicting the next 50% ahead of each model's capabilities.

$$\underset{M \in \{0,1\}^{n \times m}}{\arg\max} \mathrm{DIS}(M) = \arg\max_M \sum_{k=1}^{m} \sum_{1 \le i < j \le n} \mathbf{1}\{M_{j,k} = 1, \ M_{i,k} = 0\} \tag{8}$$

$$= \arg\max_M \sum_{1 \le i < j \le n} \left( L(j) - \sum_{k=1}^{m} \mathbf{1}\{M_{j,k} = 1, \ M_{i,k} = 1\} \right) \tag{9}$$

$$= \arg\max_M \left( \sum_{1 \le i < j \le n} L(j) \right) - \left( \sum_{k=1}^{m} \sum_{1 \le i < j \le n} \mathbf{1}\{M_{j,k} = 1, \ M_{i,k} = 1\} \right) \tag{10}$$

$$= \arg\max_M \left( \underbrace{\sum_{i=1}^{n} (i-1)L(i)}_{\text{const}} \right) - \left( \sum_{k=1}^{m} \binom{t_k}{2} \right) \tag{11}$$

$$= \arg\min_M \sum_{k=1}^{m} \binom{t_k}{2} \tag{12}$$

$$= \arg\min_M \sum_{k=1}^{m} \left( \frac{1}{2} t_k^2 \right) - \underbrace{\sum_{k=1}^{m} \left( \frac{1}{2} t_k \right)}_{\text{const}} \tag{13}$$

$$= \arg\min_M \sum_{k=1}^{m} t_k^2 \tag{14}$$

Where $t_k$ is the column sum of the $k$-th column. Thus, the matrix $M$ that maximizes $\mathrm{DIS}(M)$ and obtains $Q2_{\text{opposite}}$ is simply obtained by minimizing the sum of squares of the column sums (note that the total number of ones in the matrix is a constant decided by $\sum_{i=1}^{n} L(i)$). Since the column sums are integers and sum to a constant, and by the convexity of the square function, it is trivial to prove via contradiction (as done in A.4) that the minimizer is obtained only when all the elements are at most 1 from each other. Thus, to obtain $Q2_{\text{opposite}}$ we simply create a matrix $M$ that has row sums of $L(\cdot)$ and column sums that are different by no more than 1 from each other and compute $\mathrm{DIS}(M)$.

We construct the optimal $M$ using the following simple algorithm (Algorithm 1). Start the first row at the first column and place $L(1)$ ones contiguously to the right, wrapping around cyclically when you pass the last column. For each subsequent row $i = 2, \ldots, n$, begin one column to the right of where the previous row finished then place $L(i)$ ones contiguously with the same wraparound rule. All remaining entries are zeros. This yields a binary matrix $M$ with the required row sums which maximizes $\mathrm{DIS}(M)$.

---

**Algorithm 1** Construction of $M$ to get $Q2_{\text{opposite}}$

---

**Require:** Integers $n, m$; array $L[1..n]$ with $0 \le L[i] \le m$
**Ensure:** $M \in \{0,1\}^{n \times m}$ with row $i$ containing exactly $L[i]$ ones
 1: $M \leftarrow$ zero matrix of size $n \times m$
 2: $k \leftarrow 1$          ▷ indexing starts at 1
 3: **for** $i \leftarrow 1$ **to** $n$ **do**
 4:      **for** $r \leftarrow 1$ **to** $L[i]$ **do**
 5:          $c \leftarrow ((k-1) \bmod m) + 1$
 6:          $M[i, c] \leftarrow 1$
 7:          $k \leftarrow k + 1$
 8:      **end for**
 9: **end for**
10: **return** $M$

---

## A.4 MINIMIZER OF $\sum x_i^2$ WITH FIXED $\sum x_i$

Here we quickly prove what vector $\mathbf{x}$ minimizes $\sum x_i^2$ given a fixed $\sum x_i$ and that all $x_i$ are positive integers.

We claim that the vector $\mathbf{x}$ where all elements are at most one away from each other $((\forall\, i, j \in \{1, \ldots, n\})\, |x_i - x_j| \leq 1)$is the minimizer of $\sum x_i^2$ given a constant $\|\mathbf{x}\|_1$. We prove this is true via contradiction, assume

$$\exists\, i, j \in \{1, \ldots, n\} \text{ such that } x_i - x_j \geq 2.$$

$$\text{Define } \mathbf{y} \in \mathbb{N}^n \text{ by } \quad y_i = x_i - 1, \quad y_j = x_j + 1, \quad y_k = x_k \ (k \neq i, j).$$

(i.e. we let $\mathbf{y}$ be $\mathbf{x}$ except we bump two elements that have a big gap to be closer to each other) Then $\|\mathbf{y}\|_1 = \|\mathbf{x}\|_1$.

$$\sum_{k=1}^{n} y_k^2 = (x_i - 1)^2 + (x_j + 1)^2 + \sum_{k \neq i, j} x_k^2 = \sum_{k=1}^{n} x_k^2 - 2(x_i - x_j) + 2 \leq \sum_{k=1}^{n} x_k^2 - 2 < \sum_{k=1}^{n} x_k^2,$$

a contradiction to the minimality of $\mathbf{x}$. Thus, the vector where all elements are at most one away from each other is the minimizer of $\sum x_i^2$.

## A.5 DERIVATION OF $Q2_{\text{MATCHED}}$

Here we derive how we obtain $Q2_{\text{matched}}$ representing the theoretical maximum (perfect ordering) where higher-accuracy models perfectly subsume lower-accuracy models.

The theoretical perfect ordering is easier to compute compared to the opposite ordering (calculated in Appendix A.3), where the perfect ordering is simply when all the 1's of a column are above all the 0's in the same column. Thus for the $i$-th row of $M$ (where $M$, $L$, and $Q_3$ are defined in Appendix A.3), we simply place 1's from the first column until the $L(i)$-th column and 0's elsewhere. Since we defined the rows as ordered by non-increasing order of row sums; This construction achieves a $Q_3(i, j) = 0, \forall i, j \in \{1, \ldots, n\}\ where\ i < j$. This procedure is shown in Algorithm 2

---

**Algorithm 2** Construction of $M$ to get $Q2_{\text{matched}}$

---

**Require:** Integers $n, m$; array $L[1..n]$ with $0 \leq L[i] \leq m$
**Ensure:** $M \in \{0, 1\}^{n \times m}$ with row $i$ containing exactly $L[i]$ ones
1: $M \leftarrow$ zero matrix of size $n \times m$
2: **for** $i \leftarrow 1$ **to** $n$ **do**
3:     **for** $r \leftarrow 1$ **to** $L[i]$ **do**
4:         $M[i, r] \leftarrow 1$
5:     **end for**
6: **end for**
7: **return** $M$

---

## A.6 DERIVATION OF $Q2_{\text{RANDOM}}$

Here we provide the derivation of $Q2_{\text{random}}$ which is the value of $Q2$ across all model pairs if the predictions made by each model are independent. To calculate $Q2_{\text{random}}$ we calculate the expected value $Q2$ across all row pairs of a random matrix. We draw rows independently and uniformly from $\{0, 1\}^m$ with exactly $L(i)$ ones. For any column $k$, $\Pr[M_{i,k} = 1] = L(i)/m$ and for $i \neq j$, rows are independent at fixed $k$.

We have

$$\sum_{1 \leq i < j \leq n} Q2(i, j) = \sum_{1 \leq i < j \leq n} \sum_{k=1}^{m} \mathbf{1}\{M_{i,k} = 1,\ M_{j,k} = 0\}. \tag{15}$$

Table 6: Accuracy on Imagenet for Vision Models

|     | model | Accuracy on ImageNet |
| --- | --- | --- |
| 0 | convnext_large | 86.2% |
| 1 | convnext_base | 85.3% |
| 2 | convnext_small | 85.1% |
| 3 | swin_base_patch4_window7_224 | 84.8% |
| 4 | convnext_tiny | 84.1% |
| 5 | resnet152 | 82.6% |
| 6 | resnet101 | 82.0% |
| 7 | deit_base_patch16_224 | 81.9% |
| 8 | regnety_032 | 81.8% |
| 9 | wide_resnet50_2 | 81.5% |
| 10 | efficientnetv2_rw_m | 81.5% |
| 11 | resnext50_32x4d | 81.0% |
| 12 | vit_base_patch16_224 | 80.9% |
| 13 | swin_tiny_patch4_window7_224 | 80.9% |
| 14 | efficientnetv2_rw_s | 80.8% |
| 15 | regnety_016 | 80.6% |
| 16 | resnet50 | 80.1% |
| 17 | deit_small_patch16_224 | 79.4% |
| 18 | resnext101_32x8d | 79.1% |
| 19 | wide_resnet101_2 | 78.8% |
| 20 | regnetx_008 | 77.5% |
| 21 | densenet161 | 76.8% |
| 22 | densenet201 | 76.3% |
| 23 | resnet34 | 75.9% |
| 24 | regnety_008 | 75.8% |
| 25 | mobilenetv3_large_100 | 75.3% |
| 26 | densenet121 | 75.2% |
| 27 | densenet169 | 75.2% |
| 28 | vit_small_patch16_224 | 74.4% |
| 29 | mnasnet_100 | 74.0% |
| 30 | regnetx_006 | 73.0% |
| 31 | mobilenetv2_100 | 72.6% |
| 32 | regnetx_004 | 71.4% |
| 33 | resnet18 | 70.7% |
| 34 | inception_v3 | 69.6% |
| 35 | regnetx_002 | 67.6% |
| 36 | mobilenetv3_small_100 | 66.6% |

By linearity of expectation,

$$\mathbb{E}\left[\sum_{1 \leq i < j \leq n} Q2(i,j)\right] = \sum_{i<j} \sum_{k=1}^{m} \Pr(M_{i,k} = 1, M_{j,k} = 0) \tag{16}$$

$$= \sum_{i<j} m \cdot \frac{L(i)}{m}\left(1 - \frac{L(j)}{m}\right) \tag{17}$$

$$= \sum_{i<j} \left(L(i) - \frac{L(i)L(j)}{m}\right) \tag{18}$$

## A.7 MODELS USED FOR FUTURE PREDICTIONS

In Table 6 we list the 37 vision models which were used for the progress prediction task (depicted in Figure 6 and elsewhere) along with the validation accuracy on ImageNet:

Table 7: Accuracy on MMLU-pro for LLMs

|    | model | Accuracy on MMLU-pro |
|----|-------|----------------------|
| 0  | Qwen2.5-14B-Instruct | 52.1% |
| 1  | Phi-3-medium-4k-instruct | 47.3% |
| 2  | Qwen3-4B | 45.2% |
| 3  | Rombos-Qwen-7b | 44.6% |
| 4  | Qwen2.5-7B | 44.5% |
| 5  | Gemma-2-9B | 43.0% |
| 6  | Phi-3.5-mini | 41.2% |
| 7  | Yi-1.5-9B-Chat | 39.7% |
| 8  | Qwen2.5-3B | 37.5% |
| 9  | Mistral-Nemo-Instruct-2407 | 36.8% |
| 10 | Qwen2.5-Coder-7B-Instruct | 34.1% |
| 11 | Yi-1.5-6B-Chat | 33.1% |
| 12 | SOLAR-10.7B-Instruct-v1.0 | 31.7% |
| 13 | openchat-3.6-8b-20240522 | 31.1% |
| 14 | Mistral-7B | 30.7% |
| 15 | OpenHermes-2.5-Mistral-7B | 30.2% |
| 16 | Nous-Hermes-2-Mistral-7B-DPO | 30.0% |
| 17 | Llama-3.2-3B-Instruct | 29.1% |
| 18 | neural-chat-7b-v3-3 | 28.3% |
| 19 | zephyr-7b-beta | 28.3% |
| 20 | DeepSeek-R1-Qwen-7B | 28.1% |
| 21 | Qwen2.5-Math-7B-Instruct | 27.9% |
| 22 | Gemma-2-2B | 27.4% |
| 23 | aya-23-8B | 23.6% |
| 24 | Llama-3.2-1B-Instruct | 17.6% |
| 25 | deepseek-coder-6.7b-instruct | 16.7% |

In Table 7 we list the 26 LLMs that were used for the progress prediction task (depicted in Figure 6 and elsewhere) along with the validation accuracy on MMLU-pro (Note that the LLMs were only tasked with outputting a single token which was the answer to the multiple choice question and not given tokens to think, the prompt contained 3 shots of in context examples).

## A.8 USE OF LLMs

LLMs were used to improve the style of some of our figures. The initial figures were completely generated manually, and the LLMs improved the code for generating them while we ensured that the underlying data remained consistent and unchanged.

