# OpenReview forum: "How Predictable is AI Progress?"
_ICLR.cc/2026/Conference — Submitted to ICLR 2026_

### Official Review · Reviewer_SiLp · 2025-10-17

**Soundness:** 1
**Presentation:** 1
**Contribution:** 1
**Rating:** 2
**Confidence:** 4

**Summary:**

The paper asks whether AI capability gains are predictable with benchmark examples and introduces a new evaluation task: progress prediction, which aims to forecast which currently unsolved examples will be solved next as models improve.
To quatify the consistency of example-solving order across models, the paper introduced a metric: Prediction Order Coherence (POC).
Results across vision and language models show above-chance progress prediction across modalities and reveals that example difficulty ordering correlates with training dynamics metrics like c-score.

**Strengths:**

The paper introduces progress prediction as a new evaluation task for vision and language models.
Prediction without ground truth labels (using only model logits) is valuable for forecasting on unsolved or ambiguous problems where ground-truth is unknown or scarce.

**Weaknesses:**

1. **Progress prediction evaluated only on small models via retrospective back-testing.**
(1) The method assumes that future models will follow the same ordering as past models. But distributional shifts in training data, architecture innovations (e.g., mixture-of-experts, retrieval-augmentation), or emergent capabilities could break this assumption.
(2) The models evaluated in the experiments are in a small scale (at most 14B), thus results are unclear when transferring to larger models.
2. **POC metric assumes accuracy is a sufficient proxy for capability ordering.**
Sorting models by overall accuracy may mask capability misalignments (e.g., a lower-accuracy model excelling on a specific skill). A more nuanced ordering—e.g., by performance on skill-specific subsets—might reveal finer-grained predictability patterns.
3. **Improper and insufficient evaluation settings.**
(1) The “precision vs K/|N|” AUC metric is unconventional and might be sensitive to class imbalance and ranking shape. The binary classification variant in the appendix helps, but the main text relies on the curve AUC without clear statistical properties.
(2) LLM evaluations mix 0-shot and few-shot settings and restrict answers to single-token multiple choice. These choices can materially affect accuracy and confidence signals. Decoding temperature and prompt templates are also not detailed.
4. **Limited exploration of why ordering consistency arises.**
While the paper demonstrates that ordering is consistent and correlates with c-score, it doesn’t deeply investigate the causal mechanisms. For example, is this due to shared data priors, optimization biases, or intrinsic problem structure? A more mechanistic analysis (e.g., probing feature representations across models) would strengthen the claims.
5. **Results lack of statistical rigor and uncertainty quantification.**
Reported AUC gains over chance (e.g., 0.545–0.550 label-free predictions) are modest. No confidence intervals, bootstrapping, or statistical tests are provided to demonstrate significance and robustness.
6. **Task selection may bias results toward compositional domains.**
The paper claims in Section 7 that ordering is weaker on “trivia problems” but stronger on “compositional skills.” However, the chosen benchmarks (MATH, BBH, MUSR) are all reasoning-heavy. Including more diverse tasks (e.g., commonsense QA, creative generation) would test the generality of the claims.
7. **Lack of ablation on model population diversity.**
It’s unclear how many models are needed for a stable population ordering. Does POC saturate with 100 models, or does it keep improving with scale? An ablation study on population size and diversity (e.g., varying architecture families) would inform practical deployment.
8. **Hard to follow and understand.**
The paper lacks of many details and explanations. See questions below.

**Questions:**

1. Whar are the 1,000+vision models and 1,600+ language models used in the experiments? I only see 36 vision models and 26 language models in the appendix.
2. What does Figure 1 aim to convey?
3. Errors in Figure 4. Where is the blue dashed line? Where is random order?
4. How to see different population order difficulties in Figure 7?

---

### Official Review · Reviewer_MZXb · 2025-10-28

**Soundness:** 2
**Presentation:** 1
**Contribution:** 1
**Rating:** 2
**Confidence:** 4

**Summary:**

The paper studies the question of whether it is possible to forecast what specific elements (instances) of a benchmark will be solved by upcoming models with higher accuracy. This is connected to whether there is an overlap between the instances that a higher-accuracy model still fails to solve and those that a lower accuracy model fails to solve. They do this by considering various methods, both leveraging and ignoring knowledge of ground truth. Experiments on LLMs, vision models and protein models. They find above chance AUC when using ground truth logits, and slightly above change when not, which indicates some predictive power is possible

**Strengths:**

- I think studying how instance-level correctness evolves over models is an interesting and underaddressed question
- The finding that the difficulty prediction correlates with other difficulty metrics is interested
- the graphs showing predictions over population are insightful

**Weaknesses:**

- I think the main issue is the lack of clarity of the prediction method: the authors mention that both methods using and ignoring ground truth are used, but I am uncertain as to how these methods are actually implemented, particularly considering the different classes of models. I suggest the authors discuss this more in detail in the main body. Some of the language around this is quite handweavy. FOr instance
    - line 248 refers to “easiest K examples”: what does “easiest mean in this context
    - line 231: “what examples a model will be able to solve next if it improves its overall accuracy”: what does it mean for a model to improve its accuracy? Plausibly, the authors are talking about successive models for the same task
    - line 316 “examples that it [the model]  thinks will be solved”: how does a model think an example will be solved?
    - somewhere else, the paper mentions “population prediction order”: I am not clear if this is the same as the prediction methods used elsewhere and, if so, why are they referred to as “populational” now?

Other issues are

- Connected to the point above: the left panel of Fig 1 is also unclear, and I don’t understand the graph on the right hand side.
- unclear motivation: while it is interesting to know what tasks future model generations are going to solve, assuming that the ground truth is known makes this irrelevant to real-world applications.
- overselling the results: line 367-372 oversells the predictive power. While 0.642 AUC is quite good, this is only for cases when the ground truth is leveraged which, as I argued above, is not relevant to real-world applications (when the ground truth of a task may be unavailable). If the ground truth is not used, the AUC is only sligh6tly above chance

**Questions:**

- how is the difficulty prediction obtained?
- can the difficulty prediction be compared to baseline methods such as populational item-response theory

---

### Official Review · Reviewer_mULo · 2025-10-29

**Soundness:** 1
**Presentation:** 1
**Contribution:** 1
**Rating:** 2
**Confidence:** 5

**Summary:**

The paper introduces “progress prediction”, a task aimed at forecasting which unsolved examples current AI models will solve next as their capabilities improve. It claims that model progress follows a consistent ordering across architectures and datasets, quantified using a new metric called Prediction Order Coherence (POC).

**Strengths:**

I could not identify any clear strengths, as the paper’s main ideas and contributions are difficult to understand and insufficiently explained.

**Weaknesses:**

1. Unfulfilled claims in the introduction: The authors claim that knowing “when this will happen can be more important…” would guide research allocation, but the paper never quantifies or discusses when progress occurs, the claim remains unaddressed.

2. Vague problem framing: The text in the introduction says the problem can be viewed as an ordering problem, but it never explains what is being ordered or how the ordering is established across models, architectures, or datasets.

3. Unclear notation and definitions: In the introduction, definition of progress prediction notations such as  model, data, set of examples are all capitalized and vaguely defined. Sets are usually denoted by $\mathcal{N}$ rather than simple capital N. What are the models? They are mapping from what to what? What is the sample space of the datasets?

4. Figure organization and readability issues: Figure 1 is confusing and lacks a clear explanation of its meaning. Figures appear out of logical order (e.g., Figure 3 should precede Figure 2). All the figures are misplaced, descriptions are not in order of figure number.

5. Lack of theoretical justification for the POC metric: There is no mathematical or statistical reasoning provided to show that the Prediction Order Coherence (POC) is a valid or consistent estimator of model ordering.

6. Section 4 (“Next Solvable Example Prediction Task”) is unclear. The exposition is confusing and lacks a coherent step-by-step formulation. It is not specified whether the “lower-accuracy model” is temporally earlier or simply less capable — temporal ordering versus capability ordering is never clarified.

7. Figure 4 is ambiguous. The meaning of the overlapping red and orange lines is never explained. The figure appears disconnected from the surrounding text.

8. Overall coherence: The paper reads like a collection of loosely connected results with minimal narrative or logical flow. Core concepts (progress prediction, ordering, frontier advancement) are presented inconsistently, and no unified study emerges.

Given the unclear definitions, inconsistent figures, and absence of theoretical grounding, the submission appears premature for peer review and would require substantial restructuring before meaningful evaluation.

**Questions:**

See my above points

---

### Official Review · Reviewer_whgK · 2025-10-30

**Soundness:** 2
**Presentation:** 2
**Contribution:** 2
**Rating:** 2
**Confidence:** 4

**Summary:**

The paper claims to be able to predict capabilities of future AI

**Strengths:**

* The question is very interesting.
* There is extensive empirical analysis.

**Weaknesses:**

* The paper's scope is vague (as given by the abstract) and there is overclaiming: They defined a capability as the ability to predict on some samples, e.g., recognize some objects of images, but the term is much broader. The title is more like "Are there consistently hard and easy samples across models?"
* There are no deeper insights and incremental contrribution. The paper is poorly empirical, answering essentially the question "Are there consistently hard and easy samples across models?"   This question is not new and thus more depth would be expected. That said, I think this direction has a lot of potential as a deeper understanding of (emergent) model behavior would be great, but that would need more work.


Details:
Figure 3: on y axis, it would be nice to have sth like time or so to show the evolution across time, which would better align with your title.

**Questions:**

none

---

### Meta-Review · Area_Chair_sbir · 2025-12-04

**Summary:**

There are many reviewer concerns including:

1. Vagueness
2. Overclaiming
3. A lack of insight
4. A lack of clarity
5. Insufficient evaluation

The authors have not provided any response. All reviewers proposed reject, which I agree with.

**Reviewer Concerns:**

There is no author response, so all concerns are outstanding.

**Reviewer Scores:**

No reviewer would have changed their score as there is no author response.

---

### Decision · Program_Chairs · 2026-01-26

Reject